# A Link between Genetic Disorders and Cellular Impairment, Using Human Induced Pluripotent Stem Cells to Reveal the Functional Consequences of Copy Number Variations in the Central Nervous System—A Close Look at Chromosome 15

**DOI:** 10.3390/ijms21051860

**Published:** 2020-03-09

**Authors:** Alessia Casamassa, Daniela Ferrari, Maurizio Gelati, Massimo Carella, Angelo Luigi Vescovi, Jessica Rosati

**Affiliations:** 1Cellular Reprogramming Unit, Fondazione IRCCS Casa Sollievo della Sofferenza, Viale dei Cappuccini 1, 71013 San Giovanni Rotondo, Foggia, Italy; a.casamassa@css-mendel.it; 2Department of Environmental, Biological and Pharmaceutical Sciences and Technologies, University of Campania Luigi Vanvitelli, Viale Abramo Lincoln 5, 81100 Caserta, Italy; 3Department of Biotechnology and Biosciences, University of Milano-Bicocca, Piazza della Scienza 2, 20126 Milan, Italy; daniela.ferrari@unimib.it; 4Fondazione IRCCS Casa Sollievo della Sofferenza, Viale dei Cappuccini 1, 71013 San Giovanni Rotondo, Foggia, Italy; m.gelati@css-mendel.it (M.G.); m.carella@operapadrepio.it (M.C.)

**Keywords:** Copy Number Variation (CNV), induced pluripotent stem cells (iPSCs), 15q mice, 15q iPSCs, neurodevelopmental diseases, neuropsychiatric diseases

## Abstract

Recent cutting-edge human genetics technology has allowed us to identify copy number variations (CNVs) and has provided new insights for understanding causative mechanisms of human diseases. A growing number of studies show that CNVs could be associated with physiological mechanisms linked to evolutionary trigger, as well as to the pathogenesis of various diseases, including cancer, autoimmune disease and mental disorders such as autism spectrum disorders, schizophrenia, intellectual disabilities or attention-deficit/hyperactivity disorder. Their incomplete penetrance and variable expressivity make diagnosis difficult and hinder comprehension of the mechanistic bases of these disorders. Additional elements such as co-presence of other CNVs, genomic background and environmental factors are involved in determining the final phenotype associated with a CNV. Genetically engineered animal models are helpful tools for understanding the behavioral consequences of CNVs. However, the genetic background and the biology of these animal model systems have sometimes led to confusing results. New cellular models obtained through somatic cellular reprogramming technology that produce induced pluripotent stem cells (iPSCs) from human subjects are being used to explore the mechanisms involved in the pathogenic consequences of CNVs. Considering the vast quantity of CNVs found in the human genome, we intend to focus on reviewing the current literature on the use of iPSCs carrying CNVs on chromosome 15, highlighting advantages and limits of this system with respect to mouse model systems.

## 1. Introduction

The human genome, with its enormous structural complexity, goes through a rapid evolution due to structural variations (SV), which contribute to extend the genetic diversity among individuals and generations [1,2]. SV can include inversions and balanced translocations or genomic imbalances (insertions and deletions), which are generally found in a region of DNA measuring approximately 1 kb or longer in size. In the case of balanced rearrangements such as inversions, reciprocal translocations or copy-number-neutral insertions, there is no loss or gain of genetic material. However, when these SV determine genomic imbalances, known as copy number variants (CNVs), they alter the euploid status of DNA by changing the copy number of chromosomes or chromosomal regions.

The role of CNVs is controversial in neurodevelopmental/neuropsychiatric diseases. In particular, in prenatal diagnosis, the geneticist’s role is to determine the clinical significance [3]. The question is usually approached by analyzing whether it is a de novo mutation or whether it is inherited, assessing its size, type (deletion and duplication), gene content and number and finally, consulting previous CNV databases to ascertain whether correlations have been made between these CNVs and the development of specific diseases, such as Schizophrenia [4,5], Autism Spectrum disorders [6,7,8,9], Epileptic Encephalopathy [10] and others. For large, recurrent deletions and duplications (e.g., 16p11.2, 22q11.2, 15q13.2), the interpretation is rather obvious because there is an overwhelming genetic burden associated with the phenotypic evidence. But the cause-effect association between the presence of CNVs and a neuropsychiatric/neurodevelopmental disease is often difficult to confirm because patients carrying CNVs often present no clinical symptoms or, alternatively, may carry the same CNVs but present clinically differing pathological features [11,12]. Furthermore, CNVs with a unknown potential functional significance have also been described, known as variants of uncertain (or unknown) significance (VOUS) [13]. It is clear that there is an urgent need for a database recording all CNVs with a strong relevance in neurological disorders [14]. Likewise, clear guidelines must be established in order to distinguish disease-causing sequence variants from the many potentially functional variants present in any human genome [15], so as to avoid diagnostic errors. In the case of non-symptomatic CNVs, three possibilities must be considered: the CNV might be a benign variant, typical of a large number of individuals [16] or it could instead be pathogenic but with reduced penetrance [17,18,19] or as a third hypothesis, it might behave as a susceptibility factor which, only if combined with certain environmental conditions or a specific genetic background, might trigger the onset/progression of the disease [20,21]. In the second case, where pathological features are present, CNV could be the direct cause of disease or could play a fundamental role in its symptomatology [3].

The possible interpretations become even more complex because this genomic variability is not only present as germinal variations but also as somatic variations or somatic mosaicism [22,23], creating conditions which modify cellular genomes through ontogeny [24]. In fact, interindividual/intercellular genomic heterogeneity, both in health and disease, has often been related to somatic mosaicism [25,26]. Chromosomal mosaicism and mosaic CNVs/gene mutations are involved in brain development [26], which may have positive implications but could be also implicated in cancer [27], developmental diseases [28] and, not less significantly, may mediate neurodegeneration.

The first studies on CNVs chose an approach from a mainly genetic point of view because it was necessary to comprehend how and why a large part of chromosomes can be lost/acquired during replication [29,30,31,32] and to discriminate pathogenic or high-risk variants from benign variants in patients by quantifying the extension of heterogeneity in the human genome, in both healthy individuals and those affected by diseases [13,25]. The approach developed by the geneticists was to obtain information on CNVs from apparently healthy individuals, which could include parents, siblings and subjects from the general population. A major challenge in this field, nowadays, is to understand the role of a growing number of VOUS CNVs, which are suspected of being involved in disease susceptibility but for which additional population-level data are required [33].

Despite the ever more numerous published papers [13,34,35,36,37] showing the important role that CNVs may play in the onset/progression of several complex/genetic disorders, by far too few correlated biological studies have been published regarding the cellular implications of CNVs. Today, more data on both mouse and human models are needed in order to understand how genomic structural modifications may influence cellular behavior, causing disorders. From a biological perspective, these CNVs can cause/influence the disease phenotype through various mechanisms: (1) an alteration of copy number of dosage-sensitive genes or genes implicated in important signal transduction pathways when they are localized in coding regions [33]; (2) the interference with cis-regulatory element positioning and with the higher-order chromatin organization of the locus when they are present in non-coding regions [38,39] (Figure 1).

The aim of the first part of this review is to illustrate the complexity of this topic, taking into consideration cerebral disorders associated to CNVs of 15q chromosome and, in the second part, to show how studies on induced pluripotent stem cells (iPSCs), compared to mouse models, are increasing our comprehension of certain roles played by CNVs in cellular homeostasis. The decision to narrow down the field of description to chromosome 15 alone was made because our objectives were to highlight the pros and cons of the various model systems which have been used up to today and to illustrate how the prospects of dealing with diseases associated with CNVs are changing. Chromosome 15 allowed us to focus on a restricted area with a very wide range of possible CNVs.

## 2. The Role of CNVs in Neuropsychiatric and Neurodevelopmental Diseases

The study of the role of CNVs in disease pathogenesis is fairly recent, considering that only in the last 15 years have technologies been developed that can shed light on the relatively new concept associating the presence of CNVs to the onset/progression of complex and non-hereditary diseases [13,40,41]. Nevertheless, an increasing number of studies confirm the recurrence of specific CNVs in patients suffering from neuropsychiatric/neurodevelopmental diseases such as autism spectrum disorders, psychosis, schizophrenia and bipolar disorder [42,43,44,45,46]. As all of these diseases impact the central nervous system, understanding how these CNVs influence a complex organ like the human brain, while correlating these data to onset and disease progression, is complicated. Due to the low accessibility of brain cells, the presence of pathogenic somatically-produced CNVs in the brain has long remained “hidden.” Only recently, thanks to the use of advanced technologies such as single-cell genomics on post-mortem human tissue [26,47,48,49], findings from several studies have revealed that somatic cells in the human brain may carry genomic mosaicism, including aneuploidies, aneusomies, subchromosomal CNVs and a number of other genetic variations, both in healthy and pathological conditions [50]. This phenomenon is a natural part of the brain’s developmental processes for generating neuronal phenotypic diversity [22] but could also cause brain malformations [51,52] and neurological diseases (Figure 2). When this genomic mosaicism takes place on a genetic background where other inherited CNVs (benign or VOUS) are present, the nervous system is more likely to incur in CNV-associated diseases. Moreover, one particular mutation might increase the risk of incurring in a broad range of clinical psychopathologies, thus determining a different phenotype in patients with different genetic/environmental backgrounds, associated with more than one disease. Furthermore, CNV-associated severe mental disorders are most likely to reduce fecundity in the patient. Thus, high penetrance CNVs might be subject to negative selection pressure, persisting as rare variants and remaining undetected through genetic correlation studies.

Some of the syndromes directly associated with CNVs include: DiGeorge/Velocardiofacial syndrome (OMIM 188400 and 192430) [53,54], Williams–Beuren syndrome (OMIM 194050) [55], Smith–Magenis syndrome (OMIM 182290) [56], Prader-Willi syndrome (OMIM 176270) and Angelman syndrome (OMIM 105830) [57]. These syndromes are characterized by recurrent CNVs, relatively frequent microdeletions or microduplications, with a high penetrance in the phenotype. On the other hand, certain CNVs have incomplete penetrance and variable expressivity (e.g., 1q21.1 [58], 15q13.3, 16p11.2 [59], 16p12.1 [60] and 16p13.11 [61], 22q11.21 [62], 22q13.3 [63] microdeletions and microduplications) and additional work is required to unravel the mechanisms that determine their pathophysiology.

In order to narrow down this vast field of investigation, this review will focus solely on CNVs located on chromosome 15 and associated diseases, with the aim of comparing the results derived from evaluation of mouse models with studies developed in iPSCs and how the latter have transformed our perspectives and our prospects for understanding these syndromes. The complexity of this region (chromosome 15) is highlighted on the one hand by the number of genotypes that have so far been characterized in patients and on the other hand, by the great variability of the correlated phenotypes. In the chromosome region 15q11-13 there are 5 break-points (BP), which can create different CNVs. Observations of this complex region must further take into account the phenomenon of imprinting. There are regions of the genome in which the presence of two maternal or two paternal chromosomal copies result in anomalies in the subjects’ physiology [64]. As of today it is known that: (1) some genes are expressed or repressed (imprinted) according to their maternal or paternal origin, (2) the imprinting is restricted to certain genomic regions, one of which is in fact located on human chromosome interval 15q11-q13; (3) parent-of-origin specific expression is not genetically determined: genomic imprinting is due to an epigenetic phenomenon caused by DNA methylation and histone modifications [65,66,67].

### 2.1. The BP1-BP2 Region, with an Approximate Length of 500 kb, Contains Four Non-Imprinted Genes

*TUBGCP5*, *NIPA1*, *NIPA2* and *CYFIP1*. *NIPA1*, *NIPA2* and *CYFIP1* are ubiquitously expressed throughout the central nervous system, while *TUBGCP5* is found mainly in the subthalamic nucleus. There is a strong clinical heterogeneity among individuals with the BP1-BP2 microdeletion/microduplication. It has been observed that healthy individuals carrying the deletion frequently report mild-to-moderate impairments in motor function and deficits across several cognitive domains, including mathematical and reading skills. Healthy individuals with the duplication, on the other hand, perform on a similar level as population control subjects [68]. A genome-wide search for CNVs led to the identification of a deletion in 15q11.2, significantly associated with schizophrenia and related psychoses [69,70]. Patients carrying the 15q11.2 (BP1–BP2) deletion show significant volume changes in white (WM) and grey brain matter (GM) [71]. In particular, the GM volume is reduced in a precise region of brain: the left fusiform gyrus and the left angular gyrus. The first is generally thought to support skilled and fluid reading and the second to support retrieval of mathematical skills. For this reason, these patients are at high risk for dyslexia and dyscalculia [72,73,74]. In BP1-BP2 region, *CYFIP1* is the principal candidate gene for causing mental misfunctioning in patients. CYFIP1 interacts with fragile X mental retardation protein (FMRP) and with the Rho GTPase Rac1 and is involved in regulating axonal and dendritic outgrowth [75].

### 2.2. Within the BP2-BP3 Interval both Deletions and Duplications have been Mapped

In particular, the 15q11–q13 deletion in the paternal allele provokes Prader-Willi Syndrome (PWS) [76], in which patients are characterized by much less-than-average height, cognitive impairment and above all, by hyperphagia-dependent obesity. In PWS genetically, individuals have a complete loss of all paternally expressed genes on 15q11-q13, such as *MKRN3*, *MAGEL2*, *NDN*, *NPAP1*, *SNURF-SNRPN*, the C/D box small nucleolar (sno) RNAs *SNORD109A*, *SNORD115* and *SNORD116* and noncoding RNA *IPW*. However, in rare clinical cases, the critical PWS genetic interval has been narrowed down to a region spanning the *SNORD116* repeat and the *IPW* gene (Imprinted in Prader Willi). The deletion of the maternal allele of the gene *UB3A*, an E3 ubiquitin ligase, which is present in the same BP2-BP3 interval, causes Angelman Syndrome (AS). This disorder is characterized by intellectual disability, ataxia and seizure [76]. *UB3A* duplication, most often maternally inherited, is one of the most frequent chromosomal aberrations, causing autism spectrum disorders (ASD) (15q11-q13 duplication syndrome, OMIM 608636). Recently, however, reports have shown that a paternally inherited duplication [77,78] can also cause additional several autism features, such as development/speech delay or mental retardation.

### 2.3. CNVs between BP3 and BP4 are Rare

Three articles [79,80,81] described patients carrying this deletion with abnormal characteristics. However, the absence of this region is not statistically significant with respect to the control population, considering that the deletion was inherited from parents with normal phenotypes and without mental impairments. The genes present in this region were found to impact the development and function of the nervous system, suggesting that they may play a role in causing abnormal phenotypes. These four genes are: (1) *Amyloid beta A4 precursor protein-binding family A member 2* (OMIM 602712), encoding a neuronal adapter protein which interacts with synaptic vesicle proteins and regulates neurite out-growth; (2) *Tight junction protein 1* (OMIM 601009), which regulates cell growth and stabilizes tight junctions, connecting them to the cytoskeleton; (3) *Necdin-like gene 2* (OMIM 608243), responsible for resolution of DNA recombination; (4) *FAM189A1*, encoding a transmembrane protein whose function is unknown.

### 2.4. Within BP4 and BP5, There Is a Region Encompassing 6 Genes

*FAN1*, *MTMR10*, *TRPM1*, *KLF13*, *OTUD7A* and *CHRNA7*, as well as *hsa-miR-211*, which is highly unstable due to having LCRs (Low-Copy Repeats). LCR are region-specific DNA blocks, usually of 10 to 300 kilobase (kb) in size, with a sequence identity greater than 95–97% [82,83]. In these regions, during meiosis or mitosis, it may occur that non-allelic copies of LCRs align themselves irregularly. This ‘misalignment’ and the subsequent cross-over between the two non-allelic copies can result in genomic rearrangements in progeny cells. Due to the LCRs, this region is prone to both deletion and duplication. The deletion can present in patients in both heterozygosity and homozygosity. The phenotypic spectrum of the heterozygous deletion is highly variable, ranging from mental retardation with dysmorphic features, neuropsychiatric disturbances with cognitive impairment and a risk factor for common epilepsies [84], to complete absence of clinical symptoms [81,85]. The homozygous 15q13.3 microdeletion syndrome presents more serious symptomatology, which is more easily recognizable because it is characterized by retinal dysfunction, muscular hypotonia, profound intellectual disability, refractory epilepsy, and, occasionally, macrocytosis [86,87,88]. Instead, duplication has been associated with Autism Spectrum Disorder [80,89], Bipolar Disorder [80,90], intellectual disability [80], developmental disorders [89], behavioral disorders [89], language impairment [89,91], attention deficit hyperactivity disorder (ADHD) [92], obsessive compulsive disorder (OCD) [89,93] and epilepsy [94]. However, given the rarity of the 15q13.3 duplication, the relationship between this CNV and psychiatric disorders requires further studies. Shinawi and collaborators determined that the BP4-BP5 critical region identifying patients carrying 680-kb deletion lies within the 1.5-Mb deletion of 15q13.3 and encompasses the entire *CHRNA7* gene and the first exon of one of the isoforms of *OTUD7A* [95,96]. This smaller deletion/duplication causes neurodevelopmental phenotypes and is associated with schizophrenia, suggesting that *CHRNA7* gene has a central role in the development of the majority of neurodevelopmental pathological symptoms.

## 3. CNV Mouse Models

Until recently, the study of human neurodevelopmental disorders has been limited by the fact that direct cellular and molecular investigation of human brain cells cannot be undertaken on living patients affected by these disorders. Post-mortem [97,98,99] and animal studies have provided substantial knowledge and important insights into human brain biology and pathology but both of these techniques have had inherent limitations. Although animal models have contributed greatly to our understanding of neurodevelopmental disorders, as they can be both genetically and pharmacologically manipulated, facilitating the examination of many aspects of neurogenesis and synaptogenesis under controlled conditions, they may not always be able to recapitulate the human phenotype, in particular when behavioral, affective and cognitive changes are the principal phenotypes of the disorder and the same limitation has also partially hindered the analysis of CNVs [100]. The first studies on CNVs were carried out on engineered mice models [101], using the most frequently occurring CNVs, thus without showing the variability of CNVs present in humans. The approach chosen was behavior analysis [102,103], based on the concept that each CNV is characterized by a specific set of phenotypes, which can be tested (if already known) or discovered in the corresponding animal models and subsequently compared to human behavior [104]. This approach is based on the hypothesis that there is translational validity between mice and humans, owing to similarities between the two systems. However, a closer look at these articles reveals that some mouse strains do not demonstrate all of the main characteristics of the disease, symptomatology is often incomplete with respect to the disease in humans and that the same CNV is variable among single mice strains, as demonstrated in mouse models of 15q13.3 microdeletion: Df[h15q13]/and D/mouse respectively [105,106]. Both mouse models had a deletion of the following genes: *Mtmr10*, *Trpm1*, *Kif13*, *Otud7a*. The difference was seen in the two external genes, *Chrna7* and *Fan1*. In the first case, the mouse deletion involved the upstream breakpoint between exon 9 and 10 of the *Chrna7* gene and a downstream breakpoint between exon 3 and 4 of the Fan1 gene. In the second mouse model, the *Chrna7* and *Fan1* genes were completely deleted. Regarding phenotypes, the Df(h15q13)/mouse phenotype, although demonstrating some similarities, differed from the D/mouse in several respects. Both mouse models had slightly lower locomotor activity under baseline conditions and resistance to seizures (either spontaneous or pharmacologically induced). However, although the Df(h15q13)/mice increased in body weight and maintained normal brain weight, the D/mice maintained normal body weight and increased in brain weight. This heterogeneity in the symptomatology among different mice strains can be explained in the following ways: firstly, considering that CNVs are created through gene recombination in a region characterized by segmental repeats, although both mice models underwent deletion of the same 7 genes, the exact location of this deletion was not the same between one strain of mice and the other, which could also have had an influence on nearby regions; secondly, different genetic backgrounds can influence the symptoms. Polygenic inheritance patterns characterizing these disorders cannot be recapitulated by changes in a single gene [107]. Furthermore, modifying a single gene on a mouse model, in a diverse context from the genetic background of an affected patient, may cause the information to be lost. An example of this is an engineered mouse with a single-gene deletion/duplication of *Chrna7*. While clinical data on human patients carrying 15q13.3 CNVs suggest that *CHRNA7* haploinsufficiency is strongly associated with neuropsychiatric and neurobehavioral phenotypes, *Chrna7* KO mice [108] have failed to replicate most symptomatology [109,110,111]. Mice carrying the *Chrna7* gene deletion have slight difficulties in performing certain cognitive tasks [111,112,113,114,115,116,117], most of which require prolonged attention. Yin and co-authors explain the discrepancy between human individuals and mouse models hypothesizing the following causes: (1) possible compensation by other nAChR (nicotinic receptor transcribed by *CHRNA7* gene) subunits, which may be more numerous in mice than in humans; (2) strain-related effects, with certain genetic modifiers necessary for phenotypic expression of *Chrna7* deletion lacking in the C57BL/6 J mice; (3) non correspondence in CHRNA7 functions or in the neural circuits affected by CHRNA7 between mice and humans; (4) the influence of other genes in the genomic locus, which may account for the phenotypes associated with 15q13.3 microdeletion syndrome. Beyond the behavioral study, KO *CHRNA7* brains exhibit structural changes in the CA1 region of the hippocampus, suggesting that the absence of α7 nAChRs on GABAergic interneurons may result in the lack of maturation of dendritic spine configuration and/or increased spine plasticity [118,119]. This dendritic impairment could potentially indicate the existence of problematics in the human brain which lead to a more evident behavioral phenotype.

Another example of this lack of correspondence between humans and mice can be seen from the numerous mouse models on PWS. In all of these cases, the high degree of complexity was due both to the numerosity of the genes and to the fact that these genes were located in the critical imprinting region (causative of the disease). Different kinds of PWS [120,121,122] and AS mice were produced by different laboratories [123,124,125,126,127,128], where mouse genomes were engineered either by deleting CNVs of variable sizes or by deleting single gene CNVs, following either maternal or paternal inheritance. The analyses of these mice highlighted important differences between imprinted regions in humans and mice, suggesting that the regions are in continuous and rapid evolution. Furthermore, some regions, while common to both humans and mice, contain genes that are unique to one or the other species. Again, gene behavior in the same chromosomal region between the two species is not always similar. For instance, human *Frat3* transposed to the mouse PWS/AS region acquired the imprinted status of the insertion site [129]. Moreover, some imprinted loci showed divergent imprinting between the two species [130,131,132]. Lastly, the insertion of human transgenes including an *SNRPN* transgene [133] and an *H19* transgene [134], into mice, failed to imprint the mouse genes. What is more, the probability that a negative regulatory element is species-specific was suggested by the observation that a human transgene containing both functional elements, the PWS-SRO and AS-SRO, expressed SNRPN following maternal inheritance [133].

These findings indicate a diverse epigenetic regulation among mice and humans in certain cases. Even AS mouse models, significant in studying this disease, showed that tissue-specificity of the transcript including *UBE3A*-ATS (UBE3A-Antisense DNA Strand) differed between humans and mice [135], indicating that the timing and mechanism of UBE3A repression may diverge between these species.

## 4. Induced Pluripotent Stem Cells for Modeling CNV

A further complexity in CNV studies is characterized by a high inter-individual variability in the expression of symptoms, which could be caused, according to the “two-hit” model description, by a secondary insult during development, resulting in a more severe clinical manifestation of symptoms [136,137]. The second hit influencing the phenotype can be caused by several factors: it could be another CNV, a gene mutation or an environmental event [138]. A further complicating factor involves the timing of the second hit, as different timings during neurodevelopment cause differing outcomes. Early hits cause more widespread abnormalities, as opposed to later hits, which cause more specific changes [139].

The observation of this second hit, which is not present in engineered mice models, has posed the problem of creating an appropriate kind of cellular model system in order to study the effects of genomic alteration. A human cellular model carrying the same genetic background as that of the affected patient is required, which is also capable of differentiating into the cell type influenced by the disease. The introduction of human induced pluripotent stem cells (iPSCs) has fulfilled all requirements for these studies. Today, this in vitro disease model system allows researchers to study the mechanisms of disease pathogenesis “in-a-dish,” while at the same time providing a platform for the development and study of drug effectiveness.

From an accurate analysis of published papers which come up from searches correlating words such as “CNV,” “deletion” and “duplication” with the term “iPSCs,” we found approximately one hundred papers, written in a time-span of ten years. Most of them (reviewed in 28) regard the production of single iPSC lines from donors carrying CNVs, which, in future studies, will become the cellular models of the related syndromes. A good number of these papers (28) are articles showing iPSCs differentiation in neural cells, used to study neurodevelopmental disorders, whose complexity is due to the interaction between genomic alteration and the environment. A smaller percentage of papers used iPSCs to study the role of genomic alteration in cardiac disorders. If we consider solely papers inherent to CNVs on chromosome 15, as many as 16 articles have been published in the last 10 years, describing a great variety of CNVs containing single and/or multiple genes. Moreover, the symptomatology of the patients providing the cells used for reprogramming is extremely variable, ranging from patients with disorders associated with autism, schizophrenia, ADHD, developmental and intellectual delay, to patients without behavioral anomalies (at the moment regarding only one apparently healthy subject (Table 1).

How has the utilization of iPSCs changed the approach to studying these genomic disorders? To begin with, unlike the studies on mouse models, each separate paper shows data on many different iPSCs obtained from a large numbers of patients, carrying a wide variety of CNVs, reflecting the carriers’ genomic variability, which in some cases is very high, yet all causative of the same disease (Figure 3, [105,106,108,111,112,114,117,118,120,121,123,124,125,126,128,140,145,146,149,154,162,172,174,175,176,179,189,190,191,192]).

An example is Germain’s paper [146], where iPSCs carried two major classes of 15 chromosomal duplications, both maternally inherited: (1) interstitial duplications (int dup(15)), resulting in tandem copies 15q11-q13.1 lying head-to-head on the same chromosome arm and (2) isodicentric chromosome 15 (idic(15)) duplications, resulting in two additional copies of 15q11-q13.1, between two centromeres, creating a supernumerary chromosome. Obviously, the symptomatology among patients was different: individuals with idic(15), who had 4 copies of 15q11-q13.1, were more severely affected than those with int dup, who had 3 copies. The experiments were designed in order to first compare the differences between two different duplications and subsequently, the differences between these duplications and the deletion of that region (15q11-q13.1), which causes another disease, Angelman’s Syndrome. Each of these experiments was performed both on iPSCs and on iPS-derived neural cells, for further comparison. The obtained results were very interesting and added new information about the regulation of CNV expression and its association with the disease. The researchers found that in iPSCs and neurons carrying either the deletion or idic (15), the overall gene expression levels of the chromosome 15q genes largely reflected the copy number. This does not occur in iPSCs and neurons carrying paternal or maternal int dup (15), suggesting that the head-to-head duplication may disrupt distal regulatory elements that play important roles in forming neural tissue. They also compared global transcriptome expression between 1511-q13 deletion and idic (15) neurons, finding that most of the genes expressed (75% of the total) shared the same direction of regulation, although the genetic anomalies were opposing (deletion vs. duplication), respectively, in AS and idic (15). In fact, downregulation influenced the genes implicated in the development of neurons, including many genes potentially involved in autism. The authors commented this result hypothesizing that total gene expression in mature neurons may be the result and not the cause of impaired neuron functionality. These data indicate that neuronal gene regulation between int dup (15) and idic (15) follow different modalities, while the neuronal pathways that are disrupted in deletions and duplication of chromosome 15q11-q13.1 are similar. In this paper [146] as in all other studies on diseases of the nervous system, the differentiation of iPSCs into neurons provided the researchers with the capacity to perform molecular and cellular analyses and to test therapeutic compounds in live human cells. They reverted the expression of one of the principal genes implicated in these syndromes: *UBE3A* in iPS-derived neurons, using pharmacological compounds [146]. The study of iPS-derived cells during differentiation enabled the researchers to demonstrate that some large CNVs influenced cellular phenotypes of neural cells, showing a cause-effect association among duplicated/deleted genes and their functions [146], changing the approach from a behavioral perspective (mouse model) to one focused on cellular function (iPSCs). Another example is the study by Yoon and collaborators [145], who established multiple iPSC lines carrying 15q11.2del which were compared to iPSCs obtained by healthy donors. They did not find consistent differences in the proliferation or in differentiation efficacy but, during differentiation, when neural rosettes were formed, they noticed that the structure of adherens junctions was disrupted, suggesting that genes present in 15q11.2 CNV might have a role in the regulation of apical polarity and in the maintenance of adherens junctions, demonstrating how *CYFIP* gene deletion influenced early mammalian development. This article provides an example of how CNVs could lead to specific cellular abnormalities which might be implicated both in the neurodevelopmental origin of these disorders and in the symptomatology. The anomalies found, which cannot be observed on a macroscopic level with imaging technologies, allowed researchers to generate new hypotheses, subsequently verifying them through gene expression analyses on post-mortem human brains or on mouse models.

The following year, Das and co-authors [149] published a paper regarding the production and differentiation of an iPSCs line carrying 15q11.2(BP1-BP2)del, observing another neural phenotype, altered dendritic morphology associated with a relative immaturity of neurons, which can be correlated with the findings in the previous article and which might be explained by signaling pathways associated precisely with *CYFIP* deletion.

Another example of a study of the mechanisms associated with the genes duplicated/deleted in the CNVs was carried out by Gillentine [162], who established iPSCs from individuals both with 15q13.3 microdeletions and microduplications, differentiating them into cortical-like neural progenitor cells (NPCs). This area includes the *CHRNA7* gene, which, in the affected population, is considered to be a strong candidate gene for many of the phenotypes observed in individuals with 15q13.3 CNVs. However, as described above, *Chrna7* knockout mice exhibited very few of the human behavioral phenotypes, as though the mice had some kind of compensatory mechanism. On the contrary, in the studies on human iPS-derived neural precursor cells (NPCs), the authors succeeded in identifying a pathogenic mechanism for both *CHRNA7* deletions and duplications, which results in a decreased α7 nAChR-dependent calcium flux. These findings on duplication, which were completely unexpected, show a very logical association with the clinical results of both groups of patients, who demonstrated a reduction in calcium flux. On the basis of this observation, the authors then attempted to identify a mechanism. Among the pathways actively influenced in response to α7 nAChR calcium influx, there is the JAK2-PI3K pathway, whose effects are anti-apoptotic, anti-inflammatory and neuroprotective. They found that in both CNV groups, JAK2 (OMIM: 147796), directly activated by α7 nAChR- dependent calcium flux, had significantly decreased expression compared to controls, with its downstream target, PI3K, having decreased mRNA expression in both groups but only significantly for deletions. The wide variety of processes that may be affected by the changes in calcium flux in the cells of probands with 15q13.3 CNVs may contribute to the variable expressivity of the phenotypes observed. As in the previous articles, the authors state that the neural cells can be grown in presence of pharmacological compounds capable of modifying calcium flux, with prospects for drug development for treatment of this CNVs, with the option of focusing on both deletion and duplication probands. Indubitably, the use of iPSCs has brought about a change in perspective: it can now be said that CNVs are constituted by sets of genes that carry out precise functions within the cell, in certain cases causing pathological cellular phenotypes that could become targets for pharmacological therapies in the future and that, in certain cases, whether the deregulation of a set of genes be a duplication or a deletion, it may have a similar effect within the cell.

One of the characteristics of iPSCs is their epigenetic erasure: the current opinion is that the cellular lineage identity reset during programming is associated with erasure of the cellular epigenetic background. This gives the cell a relatively neutral basis for effectuating the modifications required in order to differentiate into any cell type of the organism. This characteristic had become an impediment to the use of iPSCs for modelling human neurodevelopmental/neuropsychiatric disorders, which are greatly influenced by environmental factors, which modify epigenetics. However, various articles regarding iPSCs obtained from PWS and AS patients suggested that the PWS-Imprinting Center (IC) methylation imprint resisted the epigenetic erasure induced by reprogramming. Moreover, this imprint was also maintained during long-term culture of both human and murine Embryonic Stem cells [193,194,195]. Chamberlain et al. [196] demonstrated that human Angelman Syndrome iPSCs recapitulated the tissue-specific pattern of *UBE3A* imprinting. During in vitro neurogenesis of the AS iPSCs, paternal *UBE3A*-ATS was expressed, while paternal *UBE3A* was repressed. On the basis of this observation, it was possible to use this model to study the regulation of *UBE3A*-ATS processing and its effects on the chromatin structure of the paternal *UBE3A* promoter during neural differentiation. These observations have opened a field of studies which will undoubtedly give interesting findings, both on the mechanisms at the basis of human imprinting and on the regulation of the principle molecular pathways implicated in Prader-Willi and Angelman Syndromes.

One of the negative features found in all of the articles is that there are many diverse protocols for generating iPSCs, which vary in efficiency and produce heterogeneous populations. This heterogeneity could be due to the fact that iPSCs accumulate mutations in culture over time or that the different methodologies that utilize viruses can influence genome stability and genome expression. It is thus necessary, especially for CNV pathologies, which are complex both in genomic structure and in correlated symptomatology, to compare iPSCs lines obtained using the same methods. The same holds true for differentiation. Current techniques for differentiating iPSCs yield heterogeneous populations of neuronal subtypes, which may have varying roles in disease pathogenesis. In the future, differentiation protocols should be defined for the production of specific neural subtypes and researchers should use the same standardized protocols. More recently, it has become evident that the production of patient-derived iPSCs for functional CNV studies necessitates more homogeneously-sorted cohorts, clinically defined with well-characterized commonly shared clinical features. These could include age of onset, endophenotypes (i.e., neurophysiological, biochemical, endocrinological, neuroanatomical, cognitive or neuropsychological features) or pharmacological response. This division into more homogeneous groups should also be useful for reducing inter-individual variation in vitro [197].

## 5. Conclusions

In past years, a strong effort has gone into developing mice models of neurodevelopmental/neuropsychiatric disorders, in the attempt to define behavioral assays reflecting their core symptoms—anomalies in social interaction, in communication, repetitive behavior [198] and others. An increasing number of CNVs associated with disorders has been replicated in transgenic models; however, these systems have a questionable validity either by virtue of simply being non-human or because they are missing the genetic background typical of affected human subjects.

As a consequence of the revolutionary work of the Yamanaka lab, somatic cells can now be reprogrammed into pluripotent stem cells from a simple patient biopsy [199,200]. This technology therefore allows researchers to use human neurons in vitro, derived specifically from patients suffering from a specific disorder, recapitulating the exact developmental events that are abnormal during the onset and the progression of the disorder [201]. These patient-specific induced pluripotent stem cells (iPSCs) present several important advantages over other in vitro systems: (1) they maintain both the primary genetic lesion and the genetic background from human individuals with confirmed diagnoses of CNV-dependent disorders; (2) they facilitate the study of larger deletions in complex genes containing internal promoters or polygenic cases which are currently difficult to engineer using genome editing technologies; (3) they have changed the approach to studying neurodevelopmental/neurodegenerative CNV-dependent syndromes. In fact, in mouse models the effects of the CNVs were analyzed through behavioral studies associated with the anatomical aspects of the organs affected by the disease, whereas in the iPSCs, what is studied is the function of the associated genes, hence the therapy, rather than focusing on reversing behavioral aspects, will be based on targeting, reducing or increasing single proteins/enzymes/RNA which are functionally impaired because of the CNVs.

Each iPSC line reflects the genetic background of the patient and its phenotypic variability reflects the symptomatic variability of each individual. Within this complexity, the comparison of iPSCs from many different patients carrying similar CNVs enables researchers to discover the molecular characteristics in common, distinguishing them from the characteristics pertaining to the genetic background of each single patient. Furthermore, the comparison among different CNVs or among opposite CNVs (deletion and duplication), in the same chromosomal region, which cause different disorders in patients, has highlighted common genes associated with different pathological phenotypes. Moreover, through studies on neural precursor cells (NPCs) obtained from iPSCs of patients affected by various differing neurodevelopmental diseases, researchers have observed that they likely converge on the same cellular phenotype, owing to NPCs differentiation and/or proliferation. How and when NPCs divide or differentiate is likely to determine the fundamental convergence point of neurodevelopmental/neuropsychiatric diseases [202]. This new approach to conceiving neurodevelopmental diseases confirms what has been found through whole exome sequencing analyses, applied to patients with congenital brain malformations and/or intellectual disability by Karaca and collaborators [203]. In particular, they demonstrated that the genes enriched in these patients converge on three cellular processes: brain development, RNA metabolism and cytoskeletal organization. Genes associated with primary microcephaly were often differentially expressed during development, with highest expression during the early embryonic and fetal periods (*ASPM*, *WDR62*, *MCPH1*, *STIL*, *KIF23* and *TTI1*). This is consistent on a cellular level with defective neurogenesis and/or loss of NPCs, resulting in decreased brain volume. The importance of homogeneity is again demonstrated by the contrast in results with two similar large-scale genomic studies published just before Karaca [204,205]. Both studies consisted of mostly consanguineous families that presented with neurodevelopmental disorders and intellectual disabilities but without homogeneous clinical features, which led to a lack of overlapping between their results and the findings of Karaca’s study, owing to a too-wide cohort population.

In conclusion, the new concept that has emerged through studies on iPSCs, verifiable through experiments on mouse models, is the understanding that certain mechanisms are common to several different neurodevelopmental diseases caused by differing CNVs [202,206].

## Figures and Tables

**Figure 1 ijms-21-01860-f001:**
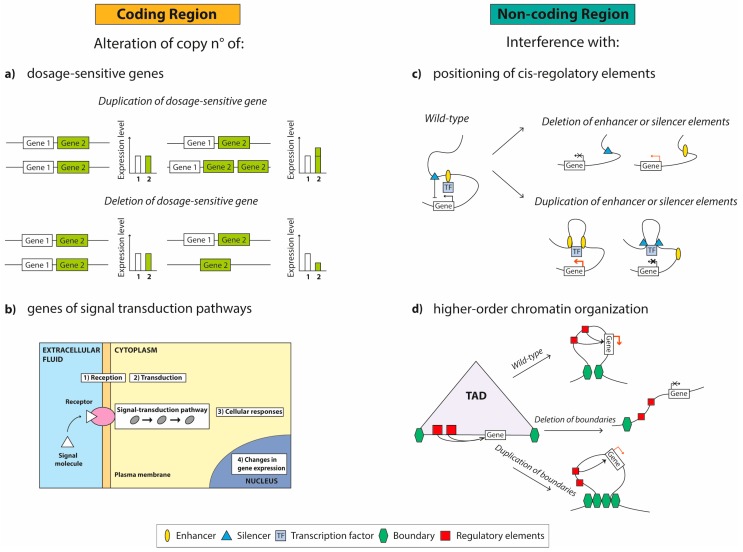
Examples of disease phenotypes caused by the presence of copy number variants (CNVs) in coding and non-coding regions. When localized in coding regions, CNVs could cause an alteration of (**a**) copy number of dosage-sensitive genes (duplications and deletions) or (**b**) genes involved in signal transduction pathways. On the other hand, CNVs in non-coding regions could interfere (**c**) with the positioning of *cis*-regulatory elements (deletion and duplication of enhancer or silencer elements) and (**d**) with the higher-order chromatin organization such as the structure of topologically associating domains (TADs) (deletion and duplication of boundaries).

**Figure 2 ijms-21-01860-f002:**
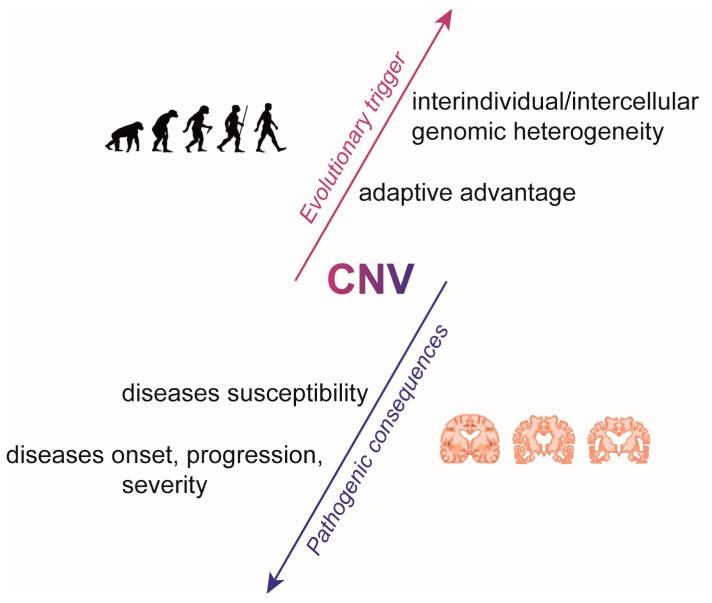
Opposite roles of CNVs: Evolution and Disease. CNVs could contribute to evolutionary trigger mechanisms as well as to the susceptibility and pathogenesis of several diseases.

**Figure 3 ijms-21-01860-f003:**
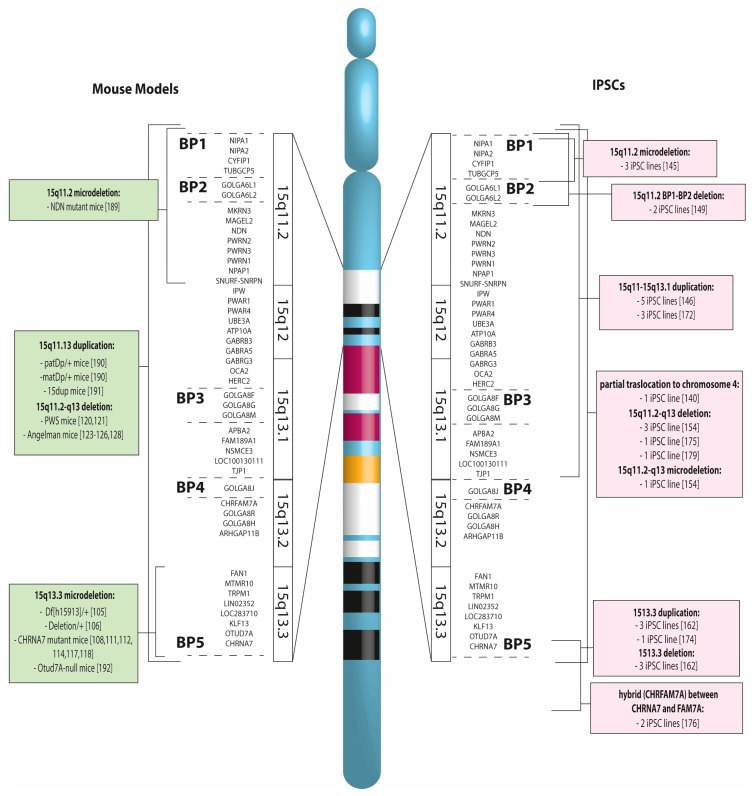
Mouse models and induced pluripotent stem cells (iPSC) lines carrying CNVs of chromosome 15q11-q13. On the left, mouse models syntenic to human chromosome 15q11-q13; on the right, iPSC lines obtained from patients carrying 15q11-q13 CNVs.

**Table 1 ijms-21-01860-t001:** Induced pluripotent stem cell (iPSC) lines generated from CNV-carrier patients published in the last 10 years.

CNV-Locus	CNV-Type	Associated Human Genes	Associated Phenotypes	Subjects (*n*)	Reference
15q11-q13	partial translocation to chromosome 4	MNKRN3, MAGEL2, Necdin, SNURF-SNRPN gene complex, SnoRNA gene cluster	PWS	1	[140]
2p25.3	deletion	MYT1L	DD, ID	1	[141]
21q21.3	duplication	APP	AD	2	[142]
Xq22.2	partial duplication	PLP1	PMD	1	[143]
12q14.2	duplication	TBK1	NTG	1	[144]
15q11.2	microdeletion	CYFIP1, NIPA1, NIPA2, TUBGCP5	SZ	3	[145]
15q11-q13.1	duplication	UBE3A, GABRB3, GABRG3, GABRA5, CYFIP1, NIPA1, NIPA2	ASD	5	[146]
Xp21	exon 44 deletion	DMD	DMD	1	[147]
7q35	exons 14-15 heterozygous deletion	CNTNAP2	SZ	1	[148]
15q11.2	BP1-BP2 deletion	CYFIP1, NIPA1, NIPA2, TUBGCP5	neurodevelopmental disorders	2	[149]
22q11.2	microdeletion	COMT, PRODH, TBX1, ZDHHC8, DGCR8	SZ	6	[150]
17q	deletion	EZH2	MDS	2	[151]
10q24.2	homozygous duplication cΔ491-496 in exon 15	HPS1	HPS type 1	1	[152]
1q32.2	CR1 CNV class 2; CR1-F/F	CR1	AD	1	[153]
1q32.2	CR1 CNV class 3; CR1-F/S	CR1	AD	1	[153]
15q11.2-q13	deletion	MNKRN3, MAGEL2, Necdin, SNURF-SNRPN gene complex, SnoRNA gene cluster	PWS	3	[154]
15q11.2-q13	microdeletion	SNOD109A, SNORD116, IPW	PWS	1	[154]
22q11.2	microdeletion	COMT, PRODH, TBX1, ZDHHC8, DGCR8	SZ	2	[155]
Xq28	deletion	MECP2	RTT	1	[156]
17q21.3	exon 17 deletion	BRCA1	Triple-negative breast cancer	1	[157]
19p13.13	deletion	CALR	AML	1	[158]
4q22.1	triplication	SNCA	PD	1	[159]
19p13.2	exon 4 homozygous deletion	LDLR	HoFH	1	[160]
16p11.2	deletion	region containing 29 genes	neurodevelopmental disorders	3	[161]
16p11.2	duplication	region containing 29 genes	neurodevelopmental disorders	3	[161]
15q13.3	heterozygous duplication	CHRNA7	DD, ID, ADHD	1	[162]
15q13.3	heterozygous duplication	CHRNA7	DD, ID, ADHD, ASD	1	[162]
15q13.3	heterozygous duplication	CHRNA7	Healthy subject	1	[162]
15q13.3	heterozygous deletion	CHRNA7	DD, ID, ASD	2	[162]
15q13.3	heterozygous deletion	CHRNA7	DD, ID	1	[162]
Xp21	exons 49–50 deletion	DMD	DMD	1	[163]
17p12	duplication	PMP22	CMT1A	2	[164]
16p12.1	homozygous deletion spanning exons 7–8	CLN3	Batten disease	1	[165]
16p12.1	heterozygous deletion spanning exons 7–8	CLN3	Batten disease	1	[165]
3p25.3	heterozygous deletion cΔ184-192	CAV3	Caveolinopathy	1	[166]
Xp21	exons 45–55 deletion	DMD	BMD	1	[167]
12p13.31	duplication	SLC2A3	ADHD	1	[168]
Xp21	exons 45–50 deletion	DMD	DMD	1	[169]
13q14.1	heterozygous deletion	RB1	Retinoblastoma	1	[170]
9q33.1	exonic deletion	ASTN2	SZ	1	[171]
15q11.2-13.1	duplication	MNKRN3, MAGEL2, Necdin, SNURF-SNRPN gene complex, SnoRNA gene cluster	15q11.2-q13.1 duplication syndrome	1	[172]
22q13	microdeletion	SHANK3	ASD	2	[173]
15q13.3	duplication	CHRNA7	Healthy subject	1	[174]
15q11.2-q13	deletion	UBEA3	AS	1	[175]
15q13-14	fusion gene	CHRFAM7A	AD	2	[176]
5p14	deletion	CTNND2	CdCS	1	[177]
5q13	deletion	SMN1	SMA	2	[178]
15q11.2-q13	deletion	MNKRN3, MAGEL2, Necdin, SNURF-SNRPN gene complex, SnoRNA gene cluster	PWS	1	[179]
3p21.31	homozygous deletion	CCR5	Resistance to HIV infection	3	[180]
7q11.22	deletion	AUTS2	DD, ASD	1	[181]
Xp21	exons 51–53 deletion	DMD	DMD	1	[182]
3p26.3	microduplication	CNTN6	DD, ID	1	[183]
10q21.1	deletion	PCDH15	BD	2	[184]
7q22.1	deletion	RELN	SZ	1	[184]
3p26.1	deletion	GRM7	ASD	1	[185]
11q22.3	homozygous deletion spanning exons 5–7	ATM	AT	1	[186]
6q26	exon 3 homozygous deletion	PRKN	PD	1	[187]
20p11.21	deletion	FOXA2	neurodevelopmental disorders	1	[188]

PWS = Prader-Willi Syndrome; Developmental Delay; ID = Intellectual Disability; AD = Alzheimer’s Disease; PMD = Pelizaeus-Merzbacher disease; NTG = Normal Tension Glaucoma; SZ = Schizophrenia; ASD = Autism Spectrum Disease; DMD = Duchenne Muscular Dystrophy; MDS = Myelodysplastic Syndrome; HPS = Hermansky-Pudlak Syndrome; RTT = Rett Syndrome; AML = Acute Myeloid Leukemia; PD = Parkinson’s Disease; HoFH = Homozygous Familial Hypercholesterolemia; ADHD = Attention Deficit/Hyperactivity Disorder; CMT1A = Charcot-Marie-Tooth; BMD = Becker Muscular Dystrophy; AS = Angelman Syndrome; CdCS = Cri du Chat Syndrome; SMA = Spinal Muscular Atrophy; BD = Bipolar Disorder; AT = Ataxia Telangiectasia.

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
