# Peer review of "A Link between Genetic Disorders and Cellular Impairment, Using Human Induced Pluripotent Stem Cells to Reveal the Functional Consequences of Copy Number Variations in the Central Nervous System—A Close Look at Chromosome 15"

_ijms, 2020, doi:10.3390/ijms21051860_

Round 1
Reviewer 1 Report
The authors present a review about the use of induced Pluripotent Stem (iPS) cells to study complex neurological diseases and their relation with copy number variations (CNV). There are major issues that impede the publication of the manuscript in this form, and the recommendation is to re-write at least some parts, add others, and fix numerous issues throughout the manuscript, especially regarding the references. Below a detailed description of major and minor issues.
Major issues.
The title and abstract are misleading. The present form of the title, as well as the content of the abstract (in particular, the last sentence), induces the reader to believe that the manuscript will discuss all the knowledge about CNV, iPS and neurological diseases. Instead, the authors focus only those related with a sub-region of chromosome 15q, while the others are only vaguely reported. In any scientific review, reported data are either supported by references or are just author’s opinion. The complete lack of adequate references inside the entire section 1 and the very limited use of additional references in the remaining part of the manuscript (most reported references are just inside Table 1, without any description/discussion/reference in the text) make the manuscript very difficult to read and understand. If the requested refs are the same reported in Table 1, nonetheless the authors must also cite them in the text, since the reader should not be forced to find an information hidden inside a 3 pages-long table! We found at least 55 points lacking adequate references. These points are reported at the end of the major points, below. The authors focus too much on few works and do not illustrate/discuss more works to compare their results, as a review should do; for example, the lines 287-316 deal with just one report. A section of conclusions is highly advised, where the authors summarize what is the state of the art and the future perspectives in the field. Also, the discussion of the differences among different protocols, describing pros and cons of each approach – possibly, inside a specific, independent, section – would increase the quality of the manuscript. At present, only vague outlines are reported on these topics, at the end of paragraph 4. The lack of an independent conclusion section makes the review somehow ‘truncated’.List of the sentences requiring references; for each, it is indicated the line and the word next to which the reference(s) should be added.
41 – CNVs 48 – disease 51 – features 52 – individuals 53 – penetrance 55 – disease 61 – disease 62 – implications 63 – neurodegeneration 64 – CNVs 70 – controls 74 – disorders 82 – (Figure 1) 85 – diseases 112 – (Figure 2) 117 – disease 118 – patient 123-126 – add refs after each disease 129 – add refs after each microdel/dup 138 – CNVs 143 – dyscalculia 148 – patients 149 – (FMRP) 149 – Rac1 150 – outgrowth 153 – Syndrome 156-158 – add refs after each gene 160 – gene 161 – interval 163 – deficits 165 – 608636) 166 – duplication 168 – retardation 170 – schizophrenia 171 – gene 172 – schizophrenia 185 – humans 190 – articles 192 – strains 199 – respects 201 – induced) 208 – symptoms 212 – CHRNA7 220 – Syndrome 225 – inheritance 244 – symptoms 246 – hit 250 – disease 258 – syndromes 261 – environment 262 – disorders 264 – genes 333 – erased 336 – articles 338 – processMinor points are listed below, indicating the manuscript line and what is requested.
30 – the use of ‘rather’ in unclear 33 – check punctuation 39 – SV is already defined in line 38 40 – substitute ‘larger’ with ‘longer’, since 1kb is not cytologically visible 43 – ‘diploid’ is not correct in that context; use ‘euploid’ 46 – substitute ‘hereditary’ with ‘inherited’ 54 – add ‘or a specific genetic background’ after the word ‘conditions’ 58 – ‘genome’ is redundant; delete 74 – change ‘be playing’ into ‘play’ 76 – change ‘comprehend’ into ‘understand’ 86 – change ‘argument’ into ‘topic’, or ‘matter’, or similar 95 – add ‘or silencer’ after the word ‘enhancer’ and update fig 1 accordingly 96 – define ‘TAD’ 104 – define ‘CNS’ 107 – change ‘single-cells genomic’ into ‘single-cell genomics’ 132-137 – this part suits better in paragraph 1 because it shows the aim of the review 139, 152, 153, 161, 164, 169,170 – avoid underlined/bold text 140 – why is ‘imprinting’ important for these genes/diseases? Pleas add at least some lines of comment, with references (this topic would justify also an independent paragraph, if the authors want to write it) 145 – delete parentheses around ref 13 155 – the use of ‘genetically’ is unclear 178-184 – too long and unclear sentence; please rephrase and divide. 217 – ‘functioning’ should be ‘function’ 232 – ‘elementi s’ should be ‘element is’ 237 – add ‘syndrome’ after ‘Angerman’ and change into ‘Angelman’ 251 – the use of quotation marks is unclear 256 and 263 – why authors choose two different time ranges, i.e, 5 and 10 years? Please explain 256 – there is ‘most of them’ but just one ref; that’s strange 266 – define ADHD 267 – no need to report refs numbers here, they are already inside tab 1 268-269 – last column of tab 1 should be ‘references’ and they should be listed as numbers 270 – the first abbreviation is missing 270-277 – this should be part of tab 1 caption 282 – these refs should be reported as numbers inside fig 3 and not here 283 – in fig 3 the extension of region 15 and sub-regions is unclear (oblique lines) 294 – add a comma after ‘and’ 298 – check word spacing after ‘interesting’ 318, 326, 340 – avoid ‘et al’; use ‘and co-workers’, ‘and collaborators’, or similar 319 – check word spacing 327 – check punctuation after ‘del’Author Response
Response to Reviewer 1
We would like to thank the reviewer for the helpful suggestions, which have been followed accurately. In particular:
1. The title and abstract are misleading. The present form of the title, as well as the content of the abstract (in particular, the last sentence), induces the reader to believe that the manuscript will discuss all the knowledge about CNV, iPS and neurological diseases. Instead, the authors focus only those related with a sub-region of chromosome 15q, while the others are only vaguely reported.
The title and the abstract have been modified as requested.
2. In any scientific review, reported data are either supported by references or are just author’s opinion. The complete lack of adequate references inside the entire section 1 and the very limited use of additional references in the remaining part of the manuscript (most reported references are just inside Table 1, without any description/discussion/reference in the text) make the manuscript very difficult to read and understand. If the requested refs are the same reported in Table 1, nonetheless the authors must also cite them in the text, since the reader should not be forced to find an information hidden inside a 3 pages-long table! We found at least 55 points lacking adequate references. These points are reported at the end of the major points, below.
All references have been included in the text.
3. The authors focus too much on few works and do not illustrate/discuss more works to compare their results, as a review should do; for example, the lines 287-316 deal with just one report.
Other works have been added and described.
4. A section of conclusions is highly advised, where the authors summarize what is the state of the art and the future perspectives in the field. Also, the discussion of the differences among different protocols, describing pros and cons of each approach – possibly, inside a specific, independent, section – would increase the quality of the manuscript. At present, only vague outlines are reported on these topics, at the end of paragraph 4. The lack of an independent conclusion section makes the review somehow ‘truncated’.
A separate conclusion has been inserted.
5. Minor points are listed below, indicating the manuscript line and what is requested.
30 – the use of ‘rather’ in unclear 33 – check punctuation 39 – SV is already defined in line 38 40 – substitute ‘larger’ with ‘longer’, since 1kb is not cytologically visible 43 – ‘diploid’ is not correct in that context; use ‘euploid’ 46 – substitute ‘hereditary’ with ‘inherited’ 54 – add ‘or a specific genetic background’ after the word ‘conditions’ 58 – ‘genome’ is redundant; delete 74 – change ‘be playing’ into ‘play’ 76 – change ‘comprehend’ into ‘understand’ 86 – change ‘argument’ into ‘topic’, or ‘matter’, or similar 95 – add ‘or silencer’ after the word ‘enhancer’ and update fig 1 accordingly 96 – define ‘TAD’ 104 – define ‘CNS’ 107 – change ‘single-cells genomic’ into ‘single-cell genomics’ 132-137 – this part suits better in paragraph 1 because it shows the aim of the review 139, 152, 153, 161, 164, 169,170 – avoid underlined/bold text 140 – why is ‘imprinting’ important for these genes/diseases? Pleas add at least some lines of comment, with references (this topic would justify also an independent paragraph, if the authors want to write it) 145 – delete parentheses around ref 13 155 – the use of ‘genetically’ is unclear 178-184 – too long and unclear sentence; please rephrase and divide. 217 – ‘functioning’ should be ‘function’ 232 – ‘elementi s’ should be ‘element is’ 237 – add ‘syndrome’ after ‘Angerman’ and change into ‘Angelman’ 251 – the use of quotation marks is unclear 256 and 263 – why authors choose two different time ranges, i.e, 5 and 10 years? Please explain 256 – there is ‘most of them’ but just one ref; that’s strange 266 – define ADHD 267 – no need to report refs numbers here, they are already inside tab 1 268-269 – last column of tab 1 should be ‘references’ and they should be listed as numbers 270 – the first abbreviation is missing 270-277 – this should be part of tab 1 caption 282 – these refs should be reported as numbers inside fig 3 and not here 283 – in fig 3 the extension of region 15 and sub-regions is unclear (oblique lines) 294 – add a comma after ‘and’ 298 – check word spacing after ‘interesting’ 318, 326, 340 – avoid ‘et al’; use ‘and co-workers’, ‘and collaborators’, or similar 319 – check word spacing 327 – check punctuation after ‘del’
All changes requested have been done.

Reviewer 2 Report
Casamassa et al. wrote the review that summarizes roles of copy number variations (CNVs), especially CNVs in human chromosome XV in cerebral disorders. This review is well written and provides an overview of how mice and iPS cells are utilized in this research field. However, it would be better to address the following concerns.
Major points:
It would be better to introduce pioneering studies that implicate CNVs in autism spectrum disorder (ASD) such as Sebat et al. (2007) Science (PMID: 17363630) and Glessner et al. (2009) Nature (PMID: 19404257). It would be better to introduce a recently published review (Hoffman et al., (2019) Molecular Psychiatry (PMID: 29483625)) that is closely related to this review, and clarify the specific focus of this review. Some sentences are too long to understand (e.g. lines 178-184; 203-208; 326-330). Please divide into short sentences to make it easier to understand. Can the authors create a figure to show the positions of breakpoints that are explained in lines 139-172. A recently published paper has demonstrated that some of the genes located in CNV regions are implicated in autism (Satterstrom et al., (2020) Cells (PMID: 31981491)). It would be better to cite this paper and introduce those genes. Please indicate the numbers of reference papers, not just the journal name and the publication year, in Table 1 to help the reader to find the papers in the list.Minor points:
Line 22-24. “However, the comprehension…” Can the authors rephrase this sentence, because it is not easy to understand. Line 29. “IPS” must be “induced pluripotent stem cells (iPSCs)” or “induced pluripotent stem (iPS) cells”. Line 33. “induced pluripotent stem cells (iPS)” must be “induced pluripotent stem cells (iPSCs)” or “induced pluripotent stem (iPS) cells”. Line 39. “Structural variation (SV)” must be “SV”. Line 73. “numerous published papers showing that ….” Please indicate the papers. Line 80 “2)” is a typo? Line 82 “Recently published papers ….” Please indicate the papers. All characters in Figure 1 are too small to read. Please increase the size of them. Lines 103-104. “The challenge of … CNS pathologies … peculiar to this field.” This sentence is unclear. Please rephrase it. It is also important to define CNS. Lines 112-114. “When this genomic mosaicism … more classical inherited CNVs … associated diseases.” This sentence is unclear. Please rephrase it. It is also unclear what “more classical inherited CNVs” mean. Line 148. “CYFP1” is “CYFIP1”? Line 150-151. “Both its deletion and duplication are associated with mental disorders.” Can the authors indicate the paper that shows the above results, and explain how both deletion and duplication of CYFIP1 might cause the disorder. Lines 162-163. “its deletion and duplication might cause neurological deficits.” What do the authors mean by “might”? Line 166. “recent papers”. Please indicate the recent papers. Lines 217-218. “in fact, nAChR … mice and humans.” This is unclear to me. Please rephrase this sentence. Line 232. “elementi s” seems to be a typo. Line 250. “induced pluripotent stem cells (iPS)” must be “induced pluripotent stem (iPS) cells” or “induced pluripotent stem cells (iPSCs)” Line 311. “In this paper”. Please indicate the paper. Line 337. Please explain “the PWS-IC methylation imprint”. Lines 360-361. Author Contribution is not explained.Author Response
Response to Reviewer 2
We would like to thank the reviewer for the careful and fair appraisal of our manuscript. All suggestions have been followed accurately. In particular:
- It would be better to introduce pioneering studies that implicate CNVs in autism spectrum disorder (ASD) such as Sebat et al. (2007) Science (PMID: 17363630) and Glessner et al. (2009) Nature (PMID: 19404257). It would be better to introduce a recently published review (Hoffman et al., (2019) Molecular Psychiatry (PMID: 29483625)) that is closely related to this review, and clarify the specific focus of this review.
All these papers have been introduced
- Some sentences are too long to understand (e.g. lines 178-184; 203-208; 326-330). Please divide into short sentences to make it easier to understand.
The sentences have been modified
- Can the authors create a figure to show the positions of breakpoints that are explained in lines 139-172. A recently published paper has demonstrated that some of the genes located in CNV regions are implicated in autism (Satterstrom et al., (2020) Cells (PMID: 31981491)). It would be better to cite this paper and introduce those genes.
The paper has been cited
- Please indicate the numbers of reference papers, not just the journal name and the publication year, in Table 1 to help the reader to find the papers in the list.
All references have been introduced
- Line 22-24. “However, the comprehension…” Can the authors rephrase this sentence, because it is not easy to understand. Line 29. “IPS” must be “induced pluripotent stem cells (iPSCs)” or “induced pluripotent stem (iPS) cells”. Line 33. “induced pluripotent stem cells (iPS)” must be “induced pluripotent stem cells (iPSCs)” or “induced pluripotent stem (iPS) cells”. Line 39. “Structural variation (SV)” must be “SV”. Line 73. “numerous published papers showing that ….” Please indicate the papers. Line 80 “2)” is a typo? Line 82 “Recently published papers ….” Please indicate the papers. All characters in Figure 1 are too small to read. Please increase the size of them. Lines 103-104. “The challenge of … CNS pathologies … peculiar to this field.” This sentence is unclear. Please rephrase it. It is also important to define CNS. Lines 112-114. “When this genomic mosaicism … more classical inherited CNVs … associated diseases.” This sentence is unclear. Please rephrase it. It is also unclear what “more classical inherited CNVs” mean. Line 148. “CYFP1” is “CYFIP1”? Line 150-151. “Both its deletion and duplication are associated with mental disorders.” Can the authors indicate the paper that shows the above results, and explain how both deletion and duplication of CYFIP1 might cause the disorder. Lines 162-163. “its deletion and duplication might cause neurological deficits.” What do the authors mean by “might”? Line 166. “recent papers”. Please indicate the recent papers. Lines 217-218. “in fact, nAChR … mice and humans.” This is unclear to me. Please rephrase this sentence. Line 232. “elementi s” seems to be a typo. Line 250. “induced pluripotent stem cells (iPS)” must be “induced pluripotent stem (iPS) cells” or “induced pluripotent stem cells (iPSCs)” Line 311. “In this paper”. Please indicate the paper. Line 337. Please explain “the PWS-IC methylation imprint”. Lines 360-361. Author Contribution is not explained.
All queries have been addressed
Reviewer 3 Report
This review is well written, compactly summarizing the impact of CNVs on the development/progression of genetic diseases especially in the central nervous system. It also illustrates the usefulness of human iPS cells in analyzing pathological bases that cannot be uncovered by using mouse models. This review will inspire the mind of young researchers who are eager to elucidate the involvement of CNVs in various human genetic disorders.
The reviewer recommends this manuscript as a strong candidate for the publication in International Journal of Molecular Science after being revised regarding the points shown below.
Minor concerns:
1) Since this review focuses on the pathological impact of CNVs in the central nervous system, it is better if the words “the central nervous system” are put into Title.
2) In lines 78-80, the phrase “… various mechanisms: on the one hand, when they are localized in coding regions, through an alteration of copy number of dosage-sensitive genes or genes implicated in important signal transduction pathways; 2) on the other hand, when they are present….” should be replaced by a clearer expression such as “… various mechanisms: 1) an alteration of copy number of dosage-sensitive genes or genes implicated in important signal transduction pathways when they are localized in coding regions ; 2)the interference with cis-regulatory element positioning and with the higher-order chromatin organization of the locus when they are presenting non-coding regions (Figure 1)”.
3) A non-abbreviated form of TAD (topologically associating domains) should be written in the legend of Figure 1.
4) Figure 1d should be replaced by a more illustrative one. The figure attached is an example.

Author Response
Response to the Reviewer 3
We would like to thank the reviewer the positive evaluation.
1. Since this review focuses on the pathological impact of CNVs in the central nervous system, it is better if the words “the central nervous system” are put into Title.
The title has been modified.
2. In lines 78-80, the phrase “… various mechanisms: on the one hand, when they are localized in coding regions, through an alteration of copy number of dosage-sensitive genes or genes implicated in important signal transduction pathways; 2) on the other hand, when they are present….” should be replaced by a clearer expression such as “… various mechanisms: 1) an alteration of copy number of dosage-sensitive genes or genes implicated in important signal transduction pathways when they are localized in coding regions; 2)the interference with cis-regulatory element positioning and with the higher-order chromatin organization of the locus when they are presenting non-coding regions (Figure 1)”.
All phrases have been modified.
3. A non-abbreviated form of TAD (topologically associating domains) should be written in the legend of Figure 1.
The non abbreviated form has been written.
4. Figure 1d should be replaced by a more illustrative one. The figure attached is an example.
The figure has been modified
Round 2
Reviewer 1 Report
The Authors significantly improved the quality and readability of the manuscript, that now is almost ready for publication. There are three major points (text needs integrations) plus some minor points (mostly typos and style issues). In the list, the first number indicates the line number of the ‘clean’ version provided.
Major points:
208 – CNVs in the BP3-BP4 interval are not described; why? ‘Rare’ means that something have been published, thus these reports should be cited and commented like for the other intervals.
210 – ‘sequence identity’; identical to what? Please specify and comment how this affects stability.
419 – ‘Figure 4D’; there is no figure 4 in the uploaded pdf file, so this figure, its caption and its use inside the text was not reviewed.
Minor points:
23 – disease should be diseases
32 – cell should be cells
51-52 – change text into ‘the geneticist has the role of determining the clinical significance’ etc.
60 – delete the underlined text in ‘often difficult’
65 – change text into ‘recording all CNVs having a strong’ etc
132 – cells should be cell
142 – ‘thirdly’ without a first and second point, is meaningless
156 – microduplication should be plural, similarly to microdeletions
160 – iPS; did the authors mean iPSCs?
174-193-208 – underlined bold text is not necessary; if the authors want to highlight the sub-paragraphs, they should use numeration (e.g.: 2.1, 2.2, etc)
193 – change text into ‘both deletions and duplications had been mapped.’
194 – no underlined text please, plus add ‘(PWS)’ after syndrome name
201 – check punctuation
202 – change provokes into causes
202 – no underlined text please
205 – no underlined text please
205 – previous OMIM numbers were without hashtag
205 – ‘reports have shown’; which reports? Add references.
210 – ‘presents’ should be changed into ‘is prone to’
218 – add ‘instead’ at the beginning of the sentence to underline that text switches from dels to dups
237 – ‘animal models’; this is duplicated with line 234; better to substitute it with ‘they’ or split sentence
250 – check punctuation after references
275 – define nAChR here and not in line 281
277 – ‘Non’ should be lowercase
277 – ‘corrispondence’ should be ‘correspondence’ (e instead of i)
328 – ‘Most of them (28)’ should be ‘Most of them (reviewed in 28)’
340 – check punctuation (fullstop, parenthesis)
371 – check punctuation (fullstop, parenthesis)
411-415-417 – a7; did the authors mean alpha7?
417 – MIM should be OMIM, similarly to lines 150-152 and 205
440 – iPS; did the authors mean iPSCs?
Author Response
We would like to thank Reviewer 1 for his/her attentive efforts in suggesting the modifications to be made in order to improve the quality of the review. We have followed all of his/her suggestions, integrating the text with the missing information and correcting the typographical errors, and hope to have succeeded in modifying the text in a satisfactory manner.
Reviewer 2 Report
Casamassa et al. addressed most of my concerns. However, I found some parts (indicated in red) are not appropriately improved in the revision. I hope the authors brush up their review article before publication.
Major points:
- It would be better to introduce pioneering studies that implicate CNVs in autism spectrum disorder (ASD) such as Sebat et al. (2007) Science (PMID: 17363630) and Glessner et al. (2009) Nature (PMID: 19404257).
- It would be better to introduce a recently published review (Hoffman et al., (2019) Molecular Psychiatry (PMID: 29483625)) that is closely related to this review, and clarify the specific focus of this review.
- Some sentences are too long to understand (e.g. lines 178-184; 203-208; 326-330). Please divide into short sentences to make it easier to understand.
- Can the authors create a figure to show the positions of breakpoints that are explained in lines 139-172.
- A recently published paper has demonstrated that some of the genes located in CNV regions are implicated in autism (Satterstrom et al., (2020) Cells (PMID: 31981491)). It would be better to cite this paper and introduce those genes.
- Please indicate the numbers of reference papers, not just the journal name and the publication year, in Table 1 to help the reader to find the papers in the list.
Minor points:
- Line 22-24. “However, the comprehension…” Can the authors rephrase this sentence, because it is not easy to understand.
- Line 29. “IPS” must be “induced pluripotent stem cells (iPSCs)” or “induced pluripotent stem (iPS) cells”.
- Line 33. “induced pluripotent stem cells (iPS)” must be “induced pluripotent stem cells (iPSCs)” or “induced pluripotent stem (iPS) cells”.
- Line 39. “Structural variation (SV)” must be “SV”.
- Line 73. “numerous published papers showing that ….” Please indicate the papers.
- Line 80 “2)” is a typo?
- Line 82 “Recently published papers ….” Please indicate the papers.
- All characters in Figure 1 are too small to read. Please increase the size of them. The characters in Fig. 1b are still small.
- Lines 103-104. “The challenge of … CNS pathologies … peculiar to this field.” This sentence is unclear. Please rephrase it. It is also important to define CNS.
- Lines 112-114. “When this genomic mosaicism … more classical inherited CNVs … associated diseases.” This sentence is unclear. Please rephrase it. It is also unclear what “more classical inherited CNVs” mean.
- Line 148. “CYFP1” is “CYFIP1”?
- Line 150-151. “Both its deletion and duplication are associated with mental disorders.” Can the authors indicate the paper that shows the above results, and explain how both deletion and duplication of CYFIP1 might cause the disorder.
- Lines 162-163. “its deletion and duplication might cause neurological deficits.” What do the authors mean by “might”?
- Line 166. “recent papers”. Please indicate the recent papers.
- Lines 217-218. “in fact, nAChR … mice and humans.” This is unclear to me. Please rephrase this sentence.
- Line 232. “elementi s” seems to be a typo.
- Line 250. “induced pluripotent stem cells (iPS)” must be “induced pluripotent stem (iPS) cells” or “induced pluripotent stem cells (iPSCs)”
- Line 311. “In this paper”. Please indicate the paper.
- Line 337. Please explain “the PWS-IC methylation imprint”.
- Lines 360-361. Author Contribution is not explained.
Author Response
We would like to thank Reviewer 2 for his/her helpful suggestions. We have inserted the missing references and have made further efforts to divide any sentences which were still a little too long into shorter ones. We have further modified Figure 1B to make it more easily legible.